# Experience with Cutaneous Manifestations in COVID-19 Patients during the Pandemic

**DOI:** 10.3390/jcm11030600

**Published:** 2022-01-25

**Authors:** Alba Navarro-Bielsa, Isabel Abadías-Granado, Ana María Morales-Callaghan, Catalina Suso-Estívalez, Marina Povar-Echeverría, Luis Rello, Yolanda Gilaberte

**Affiliations:** 1Dermatology Service, Hospital Universitario Miguel Servet, 50009 Zaragoza, Spain; isabel.abadiasg@gmail.com (I.A.-G.); ammorales@salud.aragon.es (A.M.M.-C.); ygilaberte@salud.aragon.es (Y.G.); 2IIS Aragón, Universidad de Zaragoza, 50009 Zaragoza, Spain; 3Internal Medicine Service, Hospital Universitario Miguel Servet, 50009 Zaragoza, Spain; catisuso@gmail.com (C.S.-E.); marinapovar89@hotmail.com (M.P.-E.); 4Biochemistry Service, Hospital Universitario Miguel Servet, 50009 Zaragoza, Spain; luisrello@gmail.com

**Keywords:** COVID-19, cutaneous manifestations, dermatology

## Abstract

After the beginning of the SARS-CoV-2 pandemic, our dermatology department created a multidisciplinary unit to manage patients with cutaneous manifestations associated with COVID-19. With the objective of identifying skin lesions in patients with suspected COVID-19 and evaluating possible associations with systemic involvement, other infectious agents and coagulation disorders, we carried out a prospective observational study that included all patients that attended our COVID-19 dermatology clinic with a multidisciplinary protocol. A total of 63 patients (mean 34.6 years) were enrolled between May 2020 and February 2021. Overall, 27 patients (42.9%) had a positive COVID-19 test, and 74.6% had COVID-19 clinical signs. The most common skin lesion was maculopapular rash (36.5%), predominantly seen in male (54.2%) and older patients (42 vs. 30 years), followed by chilblain-like lesions (20.6%) in younger patients (13.9 vs. 20.9 years) who were predominantly barefoot at home (69.2%); these patients exhibited a tendency towards a negative COVID-19 test. A total of 12 patients (19.1%) had positive serology for herpesvirus 6 (IgM or IgG). We conclude that the COVID-19-associated skin lesions we observed were similar to those previously described. Questions as to the underlying mechanisms remain. Interferon, possibly aided by cold exposure, may cause perniosis-like lesions. Other cutaneous manifestations were similar to those caused by other viruses, suggesting that SARS-CoV-2 may reactivate or facilitate other viral infections.

## 1. Introduction

Throughout the recent coronavirus disease 2019 (COVID-19) pandemic, several publications warned of possible cutaneous manifestations associated with this novel coronavirus (SARS-CoV-2). Galván Casas et al. [1] described the following five cutaneous clinical patterns in SARS-CoV-2 patients: acral areas of erythema with vesicles or pustules (pseudo-chilblain), other vesicular eruptions, urticarial lesions, maculopapular eruptions, and livedo or necrosis. Since then, dermatologists have reported multiple cases of patients with cutaneous clinical signs associated with COVID-19, requiring rapid scheduling of specific consultations carried out in a manner safe for all involved. In our hospital, we created a multidisciplinary unit with a specific protocol for patients with these cutaneous lesions.

The main objective of the present study was to investigate epidemiological, clinical, analytical, and microbiological factors associated with the different cutaneous manifestations of COVID-19. The secondary objective was to describe the frequency of the different cutaneous patterns observed in COVID-19 patients throughout the pandemic.

## 2. Participants and Methods

### 2.1. Study Population

An observational prospective study of patients with clinically suspected and/or microbiologically diagnosed COVID-19 and skin lesions was conducted between 1 May 2020 and 17 February 2021. A specific consultation service was created in the Department of Dermatology of the Miguel Servet University Hospital (Zaragoza, Spain) in order to attend patients referred from primary care and emergency departments, in addition to hospitalized patients and health workers. After evaluation, patients were invited to participate in the study, for which written informed consent was required.

The following inclusion criteria, similar to previous reports [1], were applied: patients of any age with clinically suspected and/or microbiologically confirmed SARS-Cov2 infection, with recent skin rash (onset within the last 4 weeks). In patients aged < 18 years, parents were required to provide informed consent and complete a questionnaire. Patients unable to complete the questionnaire or who refused to undergo a dermatological examination were excluded from the study population.

### 2.2. Questionnaire and Physical Examination

A multidisciplinary protocol was developed, including clinical assessment and completion of two questionnaires (initial consultation and after 4 weeks of follow-up. Appendix A) and a skin biopsy, if necessary. Two specific analytical profiles were created, depending on the type of cutaneous lesions: both included serology for other viruses (human herpesvirus 6 and parvovirus B19 in all cases and varicella zoster virus (VZV), syphilis, cytomegalovirus, and Epstein–Barr virus depending on the clinical manifestations), a general biochemical workup (ferritin, C-reactive protein, liver and kidney profile, vitamin D, D-dimer), and a hemogram; while the second, which was applied to patients with chilblain-like and necrotic/livedoid patterns, included an additional coagulation profile (prothrombin activity, lupus anticoagulant, anticardiolipin, anti-β2-glycoprotein, antithrombin III, protein C, protein S, and homocysteine).

### 2.3. Statistical Analyses

Qualitative variables are presented as proportions and quantitative variables as measures of central tendency (mean or median) and dispersion (standard deviation or percentiles), depending on the results of the Kolmogorov–Smirnov test.

Qualitative variables were compared using Pearson’s Chi-squared test, and the relationship between quantitative variables was compared using the Pearson or Spearman correlation. Comparison of qualitative and quantitative variables was conducted using the Student’s T-test or Mann–Whitney U-test for variables with two categories, and ANOVA or the Kruskal–Wallis test for variables with more than two categories. Parametric or nonparametric statistical tests were used as appropriate after assessing data distribution. Logistic regression was used to identify variables associated with different cutaneous manifestations. Crude odds ratios (OR) and 95% confidence intervals (CI95%) were estimated. The threshold for statistical significance was set at *p* < 0.05. Analyses were conducted using SPSS version 23.0 (IBM, Armonk, NY, USA).

### 2.4. Ethical Concerns

This study was strictly observational and the protocol was approved by the Aragón Ethical Committee for Clinical Research (CP-CI PI20/323).

## 3. Results

The characteristics of the study population are presented in Table 1. The cohort included 63 patients (61.9% female), with a mean (SD) age of 34.6 ± 21.8 years (range, 0.5–74 years). Overall, 24 patients (38.1%) exercised regularly; 16 (25.4%) had been exposed to the sun during the month preceding lesion appearance; 21 (33.3%) were usually barefoot at home; and 9 (14.3%) were smokers (5 smoked more than 10 cigarettes/day). For all analytical variables, the mean was within the normal range of our laboratory, with the exception of vitamin D levels, for which the mean (SD) value was below the normal range (67.7 (26.9) nmol/L). Serology for other viruses indicated the presence of IgM and/or IgG seroconversion to HHV-6 in 12 of the 63 patients (mean age, 30 (3–50)). The remaining 51 patients were negative for HHV-6 IgM. Of these 12 patients, only 4 (33%) tested positive for SARS-CoV-2. Results of tests for all other viruses were negative or consistent with past infection.

### 3.1. COVID-19 Clinical Signs

Most patients (74.6%) had clinical signs suggestive of COVID-19 (Table 2). The most frequent sign was fever (55.6%), followed by general malaise and/or asthenia (44.4%), headache (34.9%), digestive signs (vomiting, diarrhea, and/or abdominal pain) (28.6%), cough (28.6%), dyspnea (25.4%), and anosmia and/or dysgeusia (22.2%). Despite the high prevalence of these signs, only 42.9% had a positive COVID-19 test result (nasopharyngeal smear + PCR, 23; IgG serology, 22; IgM serology, 9), and 30 patients (47.6%) had been exposed to an individual diagnosed with COVID-19 or with suggestive signs.

### 3.2. COVID-19 Cutaneous Manifestations

The cutaneous manifestations of the cohort are summarized in Table 3. The most common were maculopapular eruptions (38.1%), followed by acral areas of erythema with vesicles or pustules (pseudo-chilblain) (20.6%), vesicular eruptions (12.7%), urticarial lesions (9.5%), and livedo or necrosis (7.9%). In patients with maculopapular eruptions, the most common pattern was morbilliform, followed by pityriasis rosea-like and eczematous, with toxicoderma suspected in 3 patients. Cutaneous signs were detected in 68.9% of participants, the most common of which were pruritus, followed by stinging and pain. None of the participants reported photosensitivity.

A total of 65% of participants received treatment, most with topical corticosteroids (44.4%), followed by oral antihistamines (25.4%) and oral corticosteroids (15.4%). After 1 month of follow-up, the cutaneous lesions had completely resolved in 52.4% of the patients, partially improved in 34.9%, and remained unchanged in 7.9%.

A total of 9 biopsies were performed, 6 in patients with maculopapular eruption, 2 with livedo and 1 with vesicular eruption; the most frequent pattern was a vacuolar and/or lichenoid superficial perivascular dermatitis, accompanied by epidermal hyperplasia consistent with drug/viral exanthematous dermatitis; dermatitis with minimal changes such as mild edema in the dermis and dilation/ectasia of superficial plexus capillaries in the livedoid pattern; and epidermal necrosis with reepithelialization probably due to subepidermal blister consistent with erythema multiforme in vesicular eruption.

The number of consultations decreased throughout the course of the pandemic Appendix A. In Aragon, 4 waves occurred during the study period. As shown in Figure 1, the frequency of the different types of dermatosis associated with COVID-19 changed with each wave, although the maculo-papular pattern was the most common in all cases. In the second wave (July–August 2020), the vesicular pattern was the second most frequent, whereas the pseudo-chilblain pattern was more frequent in the first wave (March–April 2020). Other types of cutaneous manifestations not described during the first wave [2] became more common as the pandemic progressed, especially recurrent chilblains and persistent skin manifestations, with a positive lupus anticoagulant test observed in 2 patients with post-COVID syndrome and telogen effluvia observed in 2 cases with an earlier onset than classic telogen effluvia, 2 and 4 months after the COVID-19 disease, in agreement with Rossi et al.’s description [3].

### 3.3. Bivariable and Multivariable Analysis

Table 4 shows the bivariable analysis performed to identify variables associated with any of the different cutaneous patterns. All variables for which the bivariable analysis revealed a significant association were included in the multivariate analysis.

The multivariate analysis revealed that patients with maculopapular eruptions were predominantly male (54.2% vs. 28.2% female; *p* = 0.03; OR 0.02; CI95% 1.23–18.35), were not barefoot at home (20 vs. 4; *p* = 0.02; OR 0.04; CI95% 0.31–0.94), and showed a trend towards older age (42 vs. 30 years; *p* = 0.04; OR 0.16; CI95% 0.99–1.06). Moreover, higher levels of glucose (93.8 vs. 87.7 g/L; *p* = 0.01) and GGT (29.3 vs. 20.8 U/L; *p* = 0.03) were recorded in these patients (Nagelkerke R^2^ = 0.343; Hosmer–Lemeshow test, *p* = 0.208).

Patients with the pseudo-chilblain pattern were significantly younger (13.9 vs. 20.9 years; *p* < 0.001; OR 0.01; CI95% 0.87–0.98), and 69.2% were barefoot at home (*p* < 0.001; OR 0.62; CI95% 0.27–8.71). These patients showed a trend towards a negative COVID-19 test result (11 vs. 2; OR 0.09; CI95% 0.35–1.31), and had lower levels of glucose (82 vs. 93 g/L; *p* < 0.001) and GGT (14.6 vs. 26.9 U/L; *p* = 0.01) (Nagelkerke R^2^ = 0.452; Hosmer–Lemeshow test, *p* = 0.435).

The livedo pattern was correlated with being a smoker (*p* = 0.03) and having a negative COVID-19 test (*p* = 0.035), although they were not predictive factors (OR 0.11; CI95% 0.64–55.66) and (OR 0.99; CI95% 0–13.54), respectively (Nagelkerke R^2^ = 0.306; Hosmer–Lemeshow test, *p* = 1.000).

In patients with vesicular eruption, vitamin D levels were significantly higher (94.4 vs. 63.8 nmol/L; p = 0.03) and pruritus was more frequent (7 vs. 1 patient; *p* = 0.02) (OR 0.04; CI95% 1.00–1.10) (Nagelkerke R^2^ = 0.267; Hosmer–Lemeshow test, *p* = 0.180). D-dimer levels were correlated with urticarial eruption (605.2 vs. 38 ng/mL, *p* = 0.02), but did not constitute a predictive factor (OR 0.30; CI95% 0.99–1.00) (Nagelkerke R^2^ = 0.034; Hosmer–Lemeshow test, *p* = 0.664).

## 4. Discussion

Knowledge about the cutaneous manifestations of COVID-19 is rapidly growing and has shown that many such signs can be of diagnostic and/or prognostic utility [2,4]. In fact, there are several reports warning about the appearance of similar skin manifestations in relation to COVID-19 vaccination [5]. Our study shows that although the maculopapular pattern was the most frequent COVID-19-associated pattern in our cohort, the types of lesions observed varied depending on the wave of infection, sex, age, and certain behavioral factors such as walking barefoot at home. Insufficient levels of vitamin D were associated with all cutaneous patterns except the vesicular pattern, whereas the highest D-dimer levels were observed in patients with urticaria. Finally, herpesvirus 6 IgM serology was found in 12 patients, accounting for 19% of the cohort.

The proportions and characteristics of the COVID-19-associated cutaneous manifestations in our cohort were similar to those previously described in the literature. Comparison with the findings of the Galvan et al. [1]. study reveals differences only in the case of urticarial eruption, which was recorded in 9.5% of our cohort versus 19% in that reported by Galvan et al. This discrepancy may be due to the fact that we did not attend patients in the emergency department, and recurrent urticaria and cases that are non-responsive to steroids are infrequent [6].

The age distribution was similar to that reported in other studies, and most patients presented a benign disease course. The pseudo-chilblain pattern was observed predominantly in younger patients (mean age, 13.9 years), while the remaining cutaneous manifestations were more evenly distributed. In our study, the mean (SD) age of patients with the livedo pattern was lower than that reported by Galvan and coworkers 29.1 (11.6) and 63.1 (17.3) years, respectively), and these patients had a benign course in contrast to that described by Piccolo et al. [7]. This may be due to selection bias: most patients who came to our clinic were outpatients, in contrast to the majority of other series, which consisted of hospitalized patients [2].

Only 42.9% of patients with cutaneous lesions in our cohort had a positive test of SARS-CoV-2, in line with the finding of Galvan et al. [1], whose study was conducted in similar conditions. The fact that COVID-19 tests were limited at the beginning of the pandemic likely contributed to the low percentage of confirmed cases, which subsequently increased with greater test availability.

In patients with maculopapular rash, an adverse reaction to medications prescribed for COVID-19 should be ruled out. In our cohort, 3 patients had suspected toxicoderma. In their sub-analysis of the COVID-Piel study, Català et al. reported that 78% of cases of maculopapular eruption in COVID-19 patients were associated with a history of drug use, mainly hydroxychloroquine, lopinavir/ritonavir, tocilizumab and azithromycin [8]. However, other series of cases of maculopapular rash in COVID-19 patients without a history of drug intake have also been reported [9].

The variation in the frequency of the different types of skin lesions observed over the successive waves may depend on several factors. The wide use of oral corticosteroids, not only in severe cases but also in patients with asymptomatic or mild disease [10], in addition to a decrease in the use of antimalarial and antiretroviral treatments, may contribute to a decrease in the incidence of dermatological diseases in these patients. The diagnosis of pseudo-chilblains was higher in the first wave than in the subsequent waves and was also more frequent in younger patients. This finding is in agreement with previous studies reporting an increase in cases in adults over successive waves, and in some cases, in more severe or recurrent infections [11]. Perniosis-type skin lesions or “COVID toes” are probably the most studied skin manifestations associated with COVID-19 to date and also the most controversial: in the majority of patients, SARS-CoV-2 infection could not be demonstrated, as also occurred in our cohort (15.3% positive) [12]. Some findings suggest that these lesions are a consequence of a type I interferonopathy induced by the virus [13,14], although recent publications support the lack of association between these chilblains and SARS-CoV-2 infection [15].

In our series, perniosis-like lesions were more frequent in the first wave and in children. Interestingly, we found that walking barefoot at home was significantly associated with this type of lesion, whereas no such association was observed for other patterns, such as maculopapular rash. These data support the view that lifestyle changes induced by lockdown may trigger and perpetuate the inflammatory response, as occurs in patients with type I interferonopathies [11], especially considering the improvements observed in these patients and the decrease in the incidence of perniosis like lesions after the lifting of the lockdown [16].

Laboratory tests showed that vitamin D levels were insufficient in most patients. However, we found no significant association between this parameter and the severity of cutaneous or general clinical signs of COVID-19. Interestingly, a significant association with vitamin D levels was observed only for the vesicular pattern, and these patients had vitamin D levels within the normal range. No association with previous sun exposure was observed. No studies have reported an association between vitamin D levels and any of the cutaneous manifestations included in the present study, whereas several have described an association between COVID-19 severity and vitamin D deficiency [17]. It is worth highlighting the relationship between the urticarial pattern and elevated D-dimer levels, which is considered a biomarker of disease activity and treatment response in spontaneous chronic urticaria [18].

The link between positive serology for HHV-6 and different clinical patterns in our cohort is particularly interesting and has been previously reported by our group [19]. The herpes simplex family of viruses has been implicated in both varicelliform rash [20] and pityriasis rosea-like eruptions [1,21]. In fact, several studies have reported an increase in the frequency of pityriasis rosea over the course of the pandemic [22,23] and described reactivation of the HHV-6 in cases of SARS-CoV-2 infection [24], as well as co-reactivation of HSV-1 and VZV in a patient with severe COVID-19 [25]. While it is difficult to establish an etiological association between HHV-6 infection and skin disease based on serological data alone, a growing body of evidence indicates that SARS-CoV-2 and even its vaccines [26] can reactivate other viruses [5].

The main limitation of our study is the small sample size, which was due to a gradual decrease in the number of patients referred to our clinic. This may reflect a real decrease in the incidence of COVID-19-associated cutaneous lesions, a decrease in consultations by affected patients, or an increase in knowledge by other non-dermatologists specialists about cutaneous manifestations. The other major limitation is the lack of microbiological confirmations in some of the cases, especially at the beginning of the pandemic, so we believe that it is possible that the percentage of positives would have been higher.

In conclusion, although the maculopapular pattern is the cutaneous manifestation most frequently associated with SARS-CoV-2, skin lesions vary depending on sex, age, the specific wave of the pandemic, and certain behavioral factors, such as walking barefoot at home. Further research is required to fully elucidate the role of vitamin D in the different cutaneous lesions and the relevance of the association between high D-dimer levels and the urticarial pattern. Finally, the differential diagnosis of cutaneous lesions associated with COVID-19 should include the effects of COVID-19 therapies and the reactivation of other viruses.

## Figures and Tables

**Figure 1 jcm-11-00600-f001:**
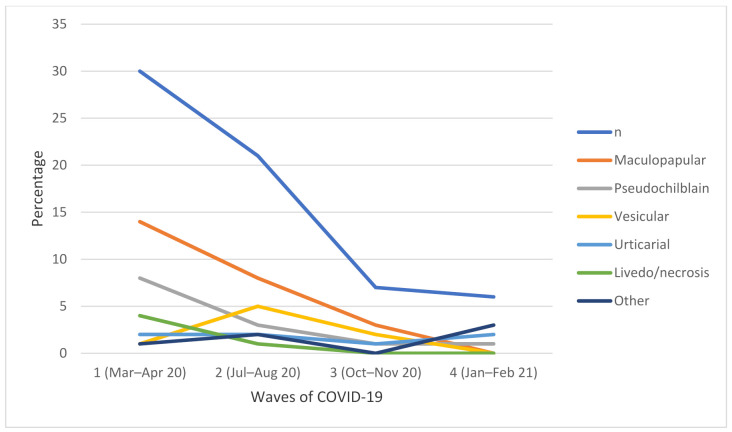
Evolution of the incidence of cutaneous lesions associated with COVID-19 over the 4 waves of the pandemic.

**Table 1 jcm-11-00600-t001:** Epidemiological, analytical, and microbiological characteristics of patients with COVID-19-associated cutaneous lesions.

Variable		
**Age,** * **mean ± SD (range)** *		34.6 ± 21.8 (0.5–74)
**Sex** * **n (%)** *	Male	24 (38.1)
	Female	39 (61.9)
**Exercise** ** *n (%)* **		24 (38.1)
**Sun exposure** ** *n (%)* **		16 (25.4)
**Barefoot at home** ** *n (%)* **		21 (33.3)
**Smokers** ** *n (%)* **		9 (14.3)
**Glucose (mg/dL),** ** *mean ± SD (range)* **		90.55 ± 12.5 (68–135)
**Urea (mg/dL),** ** *mean ± SD (range)* **		33 ± 8.4 (18–57)
**GOT (IU/L),** ** *mean ± SD (range)* **		27.2 ± 18.2 (15–152)
**GPT (IU/L),** * **mean ± SD (range)** *		27.8 ± 46.2 (6–361)
**GGT (IU/L),** ** *mean ± SD (range)* **		24.1 ± 23.6 (7–156)
**LDH (IU/L),** * **mean ± SD (range)** *		195.2 ± 51.9 (106–363)
**CRP (mg/dL),** ** *mean ± SD (range)* **		0.29 ± 0.5 (0.02–3.56)
**Ferritin (ng/dL),** * **mean ± SD (range)** *		99 ± 105.4 (10.7–553.9)
**Vitamin D (nmol/L),** * **mean ± SD (range)** *		67.7 ± 26.9 (17.9–172.1)
**Hemoglobin (g/dL),** * **mean ± SD (range)** *		13.9 ± 1,3 (10.1–16.8)
**Leucocytes** * **mean ± SD (range)** *		7149.1 ± 2619.7 (2500–16,300)
**Lymphocytes** * **mean ± SD (range)** *		2705.1 ± 1474.5 (900–11,399)
**Platelets** * **mean ± SD (range)** *		258,051 ± 76,546 (133,000–448,000)
**D-dimer (ng/mL),** * **mean ± SD (range)** *		401.4 ± 443.9 (80–3066)
**Human herpesvirus 6** * **n (%)** *	Negative	15 (23.8)
	IgG	36 (57.1)
	IgM	12 (19)
**Parvovirus B19** ** *n (%)* **	Negative	44 (69.8)
	IgG	19 (30.2)

Abbreviations: GOT, glutamic oxaloacetic transaminase; GPT, glutamic pyruvic transaminase; GGT, gamma-glutamyl transpeptidase; LDH, lactate dehydrogenase; CRP, C-reactive protein.

**Table 2 jcm-11-00600-t002:** Clinical and microbiological findings in patients with cutaneous lesions related to COVID-19 infection.

Variable	*n* (%)
COVID-19 exposure	30 (47.6)
Positive COVID-19 test	27 (42.9)
Nasopharyngeal smear + PCR	23 (36.8)
IgG serology	22 (35.2)
IgM serology	9 (14.4)
COVID-19 symptoms	47 (74.6)
Fever	35 (55.6)
General malaise/asthenia	28 (44.4)
Headache	22 (34.9)
Digestive signs	18 (28.6)
Cough	18 (28.6)
Dyspnea	16 (25.4)
Anosmia	14 (22.2)

**Table 3 jcm-11-00600-t003:** Characteristics of cutaneous lesions associated with COVID-19.

Variable	*n* (%)
Maculopapular	24 (38.1)
Morbilliform	7 (29.1)
Pityriasis rosea	5 (20.8)
Eczematous	4 (16.6)
Other	8 (33.3)
Pseudo-chilblain	13 (20.6)
Vesicular	8 (12.7)
Urticarial	6 (9.5)
Livedo/necrosis	5 (7.9)
Cutaneous signs	43 (68.3)
Pruritus	32 (74.4)
Stinging	9 (20.9)
Pain	7 (16.2)
Treatment	41 (65.1)
Topical corticosteroids	28 (44.4)
Oral antihistamines	16 (25.4)
Oral corticosteroids	10 (15.9)
Clinical skin improvement	
Complete	33 (52.4)
Partial	22 (34.9)
Null	5 (7.9)
Age, *Mean (±SD)*	
Maculopapular	42 (21.7)
Pseudo-chilblain	13.9 (9.3)
Vesicular	39.6 (22.5)
Urticarial	50.5 (21.8)
Livedo/necrosis	29.1 (11.6)

**Table 4 jcm-11-00600-t004:** Bivariable analysis performed to identify variables associated with any of the different cutaneous patterns.

Variable		Maculopapular		*p*-Value	Pseudochilblain		*p*-Value	Vesicular		*p*-Value	Urticarial		*p*-Value	Livedo/Necrosis		*p*-Value
		No	Yes		No	Yes		No	Yes		No	Yes		No	Yes	
*Mean age (SD) [range]*	34.6 (21.8) [0.5–74]	30 (20.8) [0.6–73]	42 (21.7) [2.7–74.5]	0.041	40 (20.9) [2.7–74.5]	13.9 (9.3) [0.6–34]	0.000			0.508			0.421			0.648
*Sex*	Male	11 (45.8)	13 (54.2)	0.039			0.976			0.111			0.528			0.068
	Female	28 (71.8)	11 (28.2)													
*Exercise*				0.939			0.541			0.970			0.528			0.927
*Barefoot*	No	22 (52.4)	20 (47.6)	0.028	38 (90.5)	4 (9.5)	0.002			0.789			1			0.742
	Yes	17 (81)	4 (19)		12 (57.1)	9 (42.9)										
*Smoker*	No			0.702			0.095			0.862			0.875	51 (96.2)	2 (3.8)	0.037
	Yes													7 (77.8)	2 (22.2)	
*Number of cigarettes*				0.872			0.255			0.528			0.440			0.151
*Sun previous days*				0.590			0.829			0.978			0.133			0.434
*Cutaneous signs*				0.185			0.451			0.211			0.404			0.157
*- Pruritus*	No			0.674			0.318	30 (96.8)	1 (3.2)	0.026			0.094	26 (83.9)	5 (16.1)	0.018
	Yes							25 (78.1)	7 (21.9)					32 (100)	0 (0)	
*- Stinging*	No			0.715			0.309	49 (90.7)	5 (9.3)	0.045			0.293			0.341
	Yes							6 (66.7)	3 (33.3)							
*- Pain*	No	32 (57.1)	24 (42.9)	0.028	47 (83.9)	9 (16.1)	0.011			0.285			0.363	53 (94.6)	3 (5.4)	0.032
	Yes	7 (100)	0 (0)		3 (42.9)	4 (57.1)								5 (71.4)	2 (28.6)	
*COVID-19 symptoms*				0.256			0.617			0.978			0.133			0.733
*- Fever*				0.223			0.889			0.672			0.249			0.252
*- Cough*				0.512			0.844			0.550			0.497			0.658
*- Dyspnea*				0.590			0.829			0.978			0.606			0.773
*- Myalgia*	No			0.223	24 (68.6)	11 (31.4)	0.018			0.735	34 (97.1)	1 (2.9)	0.044	30 (85.7)	5 (14.3)	0.037
	Yes				26 (92.9)	2 (7.1)					23 (82.1)	5 (17.9)		28 (100)	0 (0)	
*- Digestive signs*				0.935			0.623			0.811			0.786			0.658
*- Headache*				0.568			0.541			0.948			0.914			0.405
*- Anosmia*				0.405			0.157			0.266			0.085			0.213
*Improvement in cutaneous signs on second visit*	No			0.904			0.863			0.687			0.324	28 (84.4)	5 (15.2)	0.035
	Yes													27 (100)	0 (0)	
*Positive COVID-19 test*	No			0.525	22 (66.7)	11 (33.3)	0.015			0.135			0.261	28 (84.8)	5 (15.2)	0.035
	Yes				25 (92.6)	2 (7.4)								27 (100)	0 (0)	
*Glucose*	90.55 (12.5) [68–135]	87.7 (12.9) [68–135]	93.8 (10.8) [77–124]	0.014	93 (12) [74–135]	82 (7.8) [68–96]	0.001			0.162			0.517			0.162
*Urea*	33 (8.4) [18–57]			0.263			0.069	31.3 (9) [0.2–50]	41.4 (8.3) [31–57]	0.008			1			0.164
*GOT*	27.2 (18.2) [15–152]			0.385			0.660			0.744			0.647			0.609
*GPT*	27.8 (46.2) [6–361]			0.524			0.097			0.889			0.154			0.809
*GGT*	24.1 (23.6) [7–156]	20.8 (24.2) [7–156]	29.3 (22.1) [9–95]	0.030	26.9 (26.1) [9–156]	14.6 (5.1) [7–26]	0.019			0.102			1			0.483
*LDH*	195.2 (51.9) [106–363]			0.634			0.924			0.766			0.753			0.593
*CRP*	0.29 (0.5) [0.02–3.56]			0.209			0.459			0.271			0.224			0.060
*Ferritin*	99 (105.4) [10.7–553.9]			0.449			0.146			0.198			0.051			0.433
*Vitamin D*	67.7 (26.9) [17.9–172.1]			0.466			0.570	63.8 (22) [17.9–104.8]	94.4 (42.7) [48.7–172]	0.036			0.267			0.683
*Hemoglobin*	13.9 (1,3) [10.1–16.8]			0.179			0.097			0.907			0.830			0.500
*Leucocites*	7149.1 (2619.7) [2500–16300]			0.634			0.932			0.803			0.651			0.691
*Linfocites*	2705.1 (1474.5) [900–11399]			0.530			0.243			0.779			0.788			0.609
*Platelets*	258051 (76546) [133000–448000]			0.155			0.301			0.527			0.590			0.830
*D Dimer*	401.4 (443.9) [80–3066]			0.207			0.050			0.970	381 (452.8) [80–3066]	605.2 (304.1) [266–951]	0.026			0.113
*Herpes Virus 6*				0.748			0.310			0.916			0.490			0.388
*Parvovirus B19*				0.070			0.193			0.734			0.091			0.130
*Lupus anticoagulant positive*				0.725			0.464			0.107			0.641			0.673

Abbreviations: GOT, glutamic oxaloacetic transaminase; GPT, glutamic pyruvic transaminase; GGT, gamma glutamyl transpeptidase; LDH, lactate dehydrogenase; CRP, C-reactive protein.

## Data Availability

The data presented in this study are available on request from the corresponding author.

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
