# Peer review of "Experience with Cutaneous Manifestations in COVID-19 Patients during the Pandemic"

_jcm, 2022, doi:10.3390/jcm11030600_

Round 1

Reviewer 1 Report

The Authors presented a very contemporary and important issue. Although there are already several reports on dermatological manifestations of COVID-19, I believe many more are still required, coming from different geographical settings, to get the bigger picture of skin lesions characteristic for COVID. The advantage of the study is the multifactorial analysis, taking into account different factors that could affect the skin manifestations.

The main disadvantage of the study is that not every patient had the diagnosis of COVID confirmed. I do not see how we can draw conclusions from patients who did not have definitely COVID. Symptoms are not pathognomic, so there is a possibility that some patients suffered from other viral infections or even infections of different kind. The Authors took great lengths and performed tests for other viral pathogens indeed, but if so, I do not understand why not every patient had the test for COVID performed (Authors write something about tests limited availability). Or all patients did have test performed indeed but they were positive only in less than half on cases? Moreover, the Authors stated themselves that several patients had been positively tested for other viruses which could interfere with the outcomes.

I would enroll only patients with confirmed diagnosis of COVID.

Abstract should be unstructured - please correct.

The manuscript is not arranged as in JCM template - please correct.

Author Response

Dear Editor,

Thank you for giving us the opportunity to resubmit our manuscript. We are very grateful for your comments and those of the reviewers, which have strengthened our work. A point-by-point response follows, with changes highlighted in red.

Reviewer: 1

The Authors presented a very contemporary and important issue. Although there are already several reports on dermatological manifestations of COVID-19, I believe many more are still required, coming from different geographical settings, to get the bigger picture of skin lesions characteristic for COVID. The advantage of the study is the multifactorial analysis, taking into account different factors that could affect the skin manifestations.

The main disadvantage of the study is that not every patient had the diagnosis of COVID confirmed. I do not see how we can draw conclusions from patients who did not have definitely COVID. Symptoms are not pathognomic, so there is a possibility that some patients suffered from other viral infections or even infections of different kind. The Authors took great lengths and performed tests for other viral pathogens indeed, but if so, I do not understand why not every patient had the test for COVID performed (Authors write something about tests limited availability). Or all patients did have test performed indeed but they were positive only in less than half on cases? Moreover, the Authors stated themselves that several patients had been positively tested for other viruses which could interfere with the outcomes.

I would enroll only patients with confirmed diagnosis of COVID.

Abstract should be unstructured - please correct.

The manuscript is not arranged as in JCM template - please correct.

Response: Thank you for your comments. It is true that the major disadvantage of our study is that not all patients had microbiological confirmation of the SARS-CoV-2 infection. However, all patients seen in this clinic had been diagnosed with COVID-19, either due to an epidemiological environment and symptoms characteristic of the infection. Most of the articles that described the clinical cutaneous manifestations also included these type of patients (Galvan-Casas C, et al. Br J Dermatol 2020;183:71-77), in fact the most of the patients with chilblain-like lesions have negative microbiological test (Molaee H, et al. Childblain or perniosis-like skin lesions in children during the COVID-19 pandemic: a systematic review of articles. Dermatol Ther 2022;Jan 3;e15298). In addition, the shortfall in PCR tests was due to the limitation of diagnostic test at the beginning of the pandemic, which were reserved exclusively for severe patients. On the other hand, all patients underwent serology for the detection of antibodies against SARS-CoV-2 among other viruses, as stated. Taking all of this into consideration, the total of confirmed positives was 42.9%, but we believe that it the availability and sensibility of the microbiogical test would had been higher from the beginning.

Therefore, being consistent with previous publications, patients diagnosed of COVID-19 base of their clinical and epidemiological characteristics should be included.

Information has been added on page 8, lines 267-269.

“The other major limitation is the lack of microbiological confirmations in some of the cases, especially at the beginning of the pandemic, so we believe that it is possible the percentage of positives would have been higher.”

Also the following has been add in after the inclusion criteria:

, similar to previous reports3.

Abstract has been corrected and the manuscript has been modified according to JCM template.

Reviewer 2 Report

It was a pleasure to read this paper reporting on 63 patients with dermatological manifestations related to SARS-CoV-2: the article is well written, the methodology is sound and the results are clearly presented.

As we enter a third year of global pandemic, knowledge on SARS-CoV-2 and its cutaneous manifestations continues to evolve and all additions to our current understanding on this topic are very welcome. Moreover, global vaccination efforts have highlighted the occurrence of similar skin manifestations following SARS-CoV-2 vaccines and, though it remains beyond the scope of this study, the authors should briefly comment on this latest addition to the topic, for example referencing to https://doi.org/10.1111/dth.1515.

Specific comments:

Page 2 line 64: "...a skin biopsy, if necessary." The number of patients who underwent biopsy and a description of the diagnoses could provide interesting additions, as histological evidence on COVID-19 skin manifestations is still limited in the literature.

Page 5 line 140: "telogen effluvia". This manifestation has been recently investigated in a case series (https://doi.org/10.1159/000517223), highlighting an early onset of "COVID-19 TE" compared to "classic" acute TE. I suggest the authors to report the timing of onset of telogen effluvium after SARS-CoV-2 infection for their two cases, if available, and cite relevant literature.

Page 7 lines 218-219: "...in the majority of patients SARS-CoV-2 infection could not be demonstrated, as 218 also occurred in our cohort (15.3% positive)." According to the authors' results SARS-CoV-2 could be demonstrated in only 2 out of 13 subjects presenting with chilblain-like lesions. The authors should add a further statement on the possible lack of this association, which was demonstrated in recent literature (Hébert V, Duval-Modeste AB, Joly P, et al. Lack of association between chilblains outbreak and severe acute respiratory syndrome coronavirus 2: Histologic and serologic findings from a new immunoassay. J Am Acad Dermatol. 2020;83(5):1434-1436. doi:10.1016/j.jaad.2020.07.048).

Author Response

Dear Editor,

Thank you for giving us the opportunity to resubmit our manuscript. We are very grateful for your comments and those of the reviewers, which have strengthened our work. A point-by-point response follows, with changes highlighted in red. 

Reviewer: 2

It was a pleasure to read this paper reporting on 63 patients with dermatological manifestations related to SARS-CoV-2: the article is well written, the methodology is sound and the results are clearly presented.

As we enter a third year of global pandemic, knowledge on SARS-CoV-2 and its cutaneous manifestations continues to evolve and all additions to our current understanding on this topic are very welcome. Moreover, global vaccination efforts have highlighted the occurrence of similar skin manifestations following SARS-CoV-2 vaccines and, though it remains beyond the scope of this study, the authors should briefly comment on this latest addition to the topic, for example referencing to https://doi.org/10.1111/dth.15153.

Specific comments:

Page 2 line 64: "...a skin biopsy, if necessary." The number of patients who underwent biopsy and a description of the diagnoses could provide interesting additions, as histological evidence on COVID-19 skin manifestations is still limited in the literature.

Page 5 line 140: "telogen effluvia". This manifestation has been recently investigated in a case series (https://doi.org/10.1159/000517223), highlighting an early onset of "COVID-19 TE" compared to "classic" acute TE. I suggest the authors to report the timing of onset of telogen effluvium after SARS-CoV-2 infection for their two cases, if available, and cite relevant literature.

Page 7 lines 218-219: "...in the majority of patients SARS-CoV-2 infection could not be demonstrated, as 218 also occurred in our cohort (15.3% positive)." According to the authors' results SARS-CoV-2 could be demonstrated in only 2 out of 13 subjects presenting with chilblain-like lesions. The authors should add a further statement on the possible lack of this association, which was demonstrated in recent literature (Hébert V, Duval-Modeste AB, Joly P, et al. Lack of association between chilblains outbreak and severe acute respiratory syndrome coronavirus 2: Histologic and serologic findings from a new immunoassay. J Am Acad Dermatol. 2020;83(5):1434-1436. doi:10.1016/j.jaad.2020.07.048).

Response: thank you very much for your comments, we are glad you like the paper.

The reference to similar cutaneous manifestations in relation to COVID-19 vaccination has been added on page 6, lines 183-185.

“in fact, there are several reports warning about the appearance of similar skin manifestations in relation to the COVID-19 vaccination5.”

The information on the biopsies performed has been added in the Results section, page 4, lines 130-137.

“A total of 9 biopsies were performed, 6 in patients with maculopapular eruption, 2 with livedo and 1 with vesicular eruption; the most frequent pattern was a vacuolar and/or lichenoid superficial perivascular dermatitis, accompanied by epidermal hyperplasia consistent with drug/viral exanthematous dermatitis ; dermatitis with minimal changes such as mild edema in the dermis and dilation/ectasia of superficial plexus capillaries in the livedoid pattern; and epidermal necrosis with reepithelialization probably due to subepidermal blister consistent with erythema multiforme in vesicular eruption.”

Regarding telogen effluvia, the pertinent information has been added on page 5, lines 149-151.

“and telogen effluvia observed in 2 cases with an earlier onset than classic telogen effluvia, 2 and 4 months after the COVID-19 disease, in agreement with Rossi et al. description4

Finally, the reference regarding the lack of relationship between chilblain-like injuries and COVID-19 has been added on page 7, lines 232-233.

“although recent publications support the lack of the association these chilblains and SARS-CoV-2 infection16"

We hope you find our manuscript suitable for publication and look forward to hearing from you in due course.

Sincerely,

Dr. Alba Navarro-Bielsa

Round 2

Reviewer 1 Report

The Authors improved the manuscript and answered my comments.